# Evaluation of DAid^®^ Smart Socks for Foot Plantar Center of Pressure Measurements in Football-Specific Tasks: A Preliminary Validation Study

**DOI:** 10.3390/healthcare14010076

**Published:** 2025-12-27

**Authors:** Anna Davidovica, Guna Semjonova, Aleksejs Kataševs, Aleksandrs Okss, Darja Nesterovica, Signe Tomsone

**Affiliations:** 1Department of Rehabilitation, Riga Stradins University, 16 Dzirciema Street, LV-1007 Riga, Latvia; anna.davidovica@rsu.lv (A.D.); signe.tomsone@rsu.lv (S.T.); 2Institute of Mechanical and Biomedical Engineering, Riga Technical University, LV-1048 Riga, Latvia; aleksejs.katasevs@rtu.lv; 3Institute of Architecture and Design, Riga Technical University, LV-1048 Riga, Latvia; aleksandrs.okss@rtu.lv; 4Institute of Public Health, Riga Stradinš University, Kronvalda bulvāris 9, LV-1010, Riga, Latvia; darja.nesterovica@rsu.lv

**Keywords:** plantar pressure, wearable sensors, football, preliminary validity, FIFA 11+ exercises

## Abstract

**Highlights:**

**What are the main findings?**
The DAid^®^ Smart Socks demonstrated moderate-to-high correspondence with a gold-standard force platform in capturing the directional and temporal characteristics of plantar CoP during FIFA 11+ functional exercises.Absolute agreement varied across participants, with notable differences linked to foot morphology and sensor positioning inherent to the six-sensor textile design.

**What are the implication of the main findings?**
The system is suitable for applications where relative CoP dynamics are of primary interest, such as monitoring movement patterns and load-shift tendencies in functional tasks.Its portability and ease of use support field-based biomechanical assessment in youth football injury prevention task sequence, particularly when absolute spatial CoP measurement is not required.

**Abstract:**

**Background/Objectives:** Accurate plantar pressure assessment is essential for injury prevention and rehabilitation monitoring in sports. Wearable sensor technologies, such as DAid^®^ Smart Socks, offer portable, real-time biomechanical feedback and enable data collection in field conditions. However, there is limited evidence on their level of agreement with a gold standard in measuring the foot plantar center of pressure (CoP) in football-specific tasks. This study aimed to determine the preliminary validity of DAid^®^ Smart Socks compared with a gold-standard force platform in measuring plantar center of pressure (CoP) during functional football FIFA 11+ Part 2 exercises. **Methods:** Ten male volunteer youth football players (mean age 12.2 ± 0.42 years; height 158.7 ± 7.72 cm; weight 46.46 ± 8.78 kg; shoe size EU 39.8 ± 2.68) from the Latvian Football Federation Youth League participated. Eight players had right-leg dominance, two had left-leg dominance; three reported past lower-limb injuries. Plantar pressure was measured simultaneously using DAid^®^ Smart Socks and a 1.5 m entry-level force platform with a calibration factor of 3.2. Center of pressure (CoP) data from the force platform were recorded using Footscan software version 9.10.4. Participants performed two selected FIFA 11+ Part 2 exercises—a single-leg squat (unilateral) and a squat with heel raise, performed bilaterally—under standardized conditions. Each exercise was performed twice, with sock removal and reapplication between trials. Agreement between the DAid^®^ Smart Socks and the force platform was examined using waveform synchronization, root mean square error (RMSE), Bland–Altman analysis, and Lin’s Concordance Correlation Coefficient (CCC) to quantify both relative waveform correspondence and absolute CoP measurement accuracy. **Results:** Across 160 paired recordings, the DAid^®^ Smart Socks showed moderate-to-high correlation with the force platform for relative CoP dynamics, with 79% of waveforms demonstrating CCC ≥ 0.60. Absolute agreement was limited, with only 16% of recordings reaching CCC ≥ 0.90, and RMSE values ranging from 2.1 to 18.9 mm (X) and 4.3–34.2 mm (Y). **Conclusions:** DAid^®^ Smart Socks showed moderate-to-high correspondence with the force platform in capturing the directional and temporal characteristics of plantar CoP during functional football tasks, with agreement varying across individuals.

## 1. Introduction

Foot and ankle injuries are among the most common musculoskeletal problems in football, with epidemiological studies indicating that they account for approximately 17–33% of all injuries in both youth and professional players [1,2,3]. These injuries can occur acutely during contact situations or develop gradually as overuse syndromes, particularly in athletes exposed to repetitive loading, rapid changes in direction, and high-intensity running [4,5]. Overuse conditions, including plantar fasciitis, metatarsalgia, and stress fractures, have been directly associated with abnormal plantar pressure distribution and may lead to prolonged absence from training or competition if not addressed early [4,5,6]. In addition, overuse conditions such as chronic ankle instability (CAI) cause recurrent sprains, inadequate neuromuscular control, and altered plantar pressure patterns contribute to persistent mechanical and functional ankle instability, which may impair balance, reduce performance, and increase the long-term risk of re-injury [4]. In youth athletes, the risk is amplified by the combination of physical immaturity, frequent training loads, and incomplete recovery periods [7].

Plantar pressure analysis is an important tool in gait assessment, sports performance testing, and injury prevention. It provides clear information about how weight is distributed across the foot, helping clinicians, coaches, and sports scientists identify uneven loading, high-pressure points, and unusual movement patterns. This information can be used not only to diagnose existing problems but also to design targeted training or treatment plans to prevent injuries from recurring. In rehabilitation, plantar pressure measurement is useful for tracking recovery progress and checking whether an athlete is ready to return to sport [8].

Force plates are considered the gold-standard method for plantar pressure and ground reaction force assessment due to their high accuracy, high sampling frequency, and well-established validity and reproducibility [9,10]. Other commonly used systems, such as instrumented walkways and in-shoe pressure insoles [11,12], serve as practical alternatives but should not be classified as gold-standard tools. While they allow plantar loading assessment in more functional environments, they share several limitations, including calibration sensitivity, footwear fit interference, and reduced precision under dynamic conditions [11,12]. These constraints restrict their ability to capture representative foot-loading mechanics during sport-specific tasks and on-field activities, highlighting the need for valid wearable measurement solutions that overcome laboratory dependency.

Advances in wearable sensor technology have led to the development of textile-integrated pressure measurement systems that combine portability, real-time feedback, and minimal interference with natural movement [13]. Smart garments embed miniature pressure or motion sensors within clothing or footwear, enabling continuous monitoring during regular training or competition. In recent years, such systems have been applied in rehabilitation, injury prevention, and performance optimization across various sports [14]. Existing commercial smart socks—such as the Sensoria^®^ system—are already available on the market; however, these products remain relatively expensive and are primarily designed for step counting, gait tracking, or running analytics rather than precise plantar pressure evaluation. Importantly, current smart sock systems do not provide sensor configurations suitable for accurate plantar pressure or center of pressure monitoring, limiting their applicability in sports biomechanics and rehabilitation contexts [9].

The DAid^®^ Smart Socks system is textile-based wearable that integrates multiple pressure sensors into the fabric structure to measure plantar pressure distribution dynamically. It contains six manufactural imbedded textile sensors [15], which connects wirelessly to dedicated software for real-time data visualization and storage [16]. This portability and ease of use suggest potential applications not only in laboratory research but also in field-based athlete monitoring, rehabilitation programs, and injury prevention initiatives in sports such as football [16,17].

While early studies on smart garments and pressure-sensing insoles have demonstrated reliability, validity, and feasibility in evaluating movement patterns across a variety of clinical and sport-related tasks, these findings remain primarily limited to controlled, low-complexity movements such as walking or simple functional assessments. [16,18,19,20]. Eizentals et al. (2021) [16] have examined the performance of the DAid^®^ Smart Socks system and conducted a comprehensive performance evaluation of the DAid^®^ Pressure Socks System (DPSS) by comparing plantar pressure measurements with the Pedar^®^ in-shoe system during walking. Their findings about reliability demonstrated moderate-to-excellent agreement for most sensors, with intraclass correlation coefficients (ICC) above 0.75 in 60% of cases and above 0.90 in ~23% of cases, while identifying known limitations such as reduced accuracy under the medial arch and sensor-placement sensitivity. Additional reliability evidence has been shown for related DAid^®^ textile systems at Semjonova et al., 2019 [18], such as the DAid^®^ Smart Shirt, which demonstrated reliable and valid repeated measurements of upper-limb kinematics. These findings collectively support the technical feasibility of the smart textile-based motion capture solutions, the DAid^®^ Smart Socks system being one of them. Studies by Januskevica et al. (2020) [21] have validated DAid^®^ textile systems for detecting medial–lateral CoP shifts during functional tasks and Semjonova et al., 2022 demonstrated their usability and feasibility for real-time functional feedback [22]. Despite these results, existing research has primarily evaluated the system during controlled, low-complexity tasks (e.g., walking, static balance, or simple functional tests) and has not yet examined its validity during football-related neuromuscular control exercises. Existing research on DAid^®^ Smart Socks has shown promising potential for detecting medial–lateral CoP shifts and providing real-time functional feedback; however, these studies have not examined the system’s performance during structured neuromuscular control exercises relevant to football. This gap highlights the need for further validation under functional, sport-relevant conditions that challenge balance, lower-limb alignment, and controlled plantar loading. Without such validation against established gold-standard instruments, integration into evidence-based sports rehabilitation practice is premature. Examining their performance during football-specific functional exercises, would provide a relevant data for real-world applications in training and injury prevention programs.

The present study aims to evaluate the preliminary validity of the DAid^®^ Smart Socks by quantifying their level of agreement with a gold-standard force platform in measuring the position of the plantar center of pressure (CoP) during functional football injury-prevention tasks (FIFA 11+) performed by youth football players. In addition, as the DAid^®^ Smart Socks incorporate only six sensors, whose positioning and sensitivity may be affected by inter-individual differences in foot morphology, the study seeks to examine how the magnitude and consistency of this agreement vary across individual participants.

## 2. Materials and Methods

### 2.1. General Design of the Study

This study focuses on estimating the level of agreement between CoP waveforms recorded using the DAid^®^ Smart Socks (Riga, Latvia)—the device under study—and those obtained using the gold-standard force platform. Given the exploratory nature of the work, the study design intentionally restricts the number of tested combinations while still capturing key sources of variability:Participant-related variability, assessed across a small number of individual athletes;Task-related variability, evaluated using a limited subset of exercises from the broader FIFA 11+ program;Sock-position variability, investigated through a small number of repeated cycles in which the sock is removed and reapplied to capture the effect of putting-on-induced repositioning.

This design enables characterization of the main trends in agreement while keeping the experimental scope appropriate for a preliminary validation study.

### 2.2. Participants

Ten male (n = 10) youth football players from the Latvian Football Federation Youth League participated in this study. The mean age was 12.2 ± 0.42 years, mean height 158.7 ± 7.72 cm, mean body mass 46.46 ± 8.78 kg, and mean shoe size EU 39.8 ± 2.68. Eight participants had right-leg dominance and two had left-leg dominance. Inclusion criteria were as follows: (1) youth football league players aged 12–15 years, (2) ability to understand and follow instructions in Latvian, and (3) consent to participate. Exclusion criteria included the following: (1) lower-limb surgery or injury in the past six months, and (2) vestibular system disorders. All participants and their legal guardians provided written informed consent. The study was conducted in accordance with the Declaration of Helsinki, Latvian legislation, and the European Union’s General Data Protection Regulation (GDPR) 2016/679. Ethical approval was obtained from the Riga Stradins University Research Ethics Committee on 21 March 2023 (Approval No. 2-PĒK-4/294/2023).

### 2.3. Intervention

The present study focused on two functional movements taken from FIFA 11+ Part 2: (1) a unilateral single-leg squat and (2) a bilateral squat with heel raise [23,24]. The single-leg squat is performed unilaterally, whereas the squat with heel raise is executed bilaterally; this distinction is important for understanding the expected CoP behavior and load distribution and these tasks are widely used in football injury-prevention programs and screening routines [25]. Although not football-specific in the sense of replicating sport-specific actions such as cutting or sprinting, these movements are general functional exercises that challenge neuromuscular control, lower-limb alignment, and dynamic stability [26,27]. These components are highly relevant in football and are routinely included in warm-up and conditioning protocols aimed at reducing injury risk. The selected tasks provide controlled, repeatable movement patterns that allow precise comparison of CoP trajectories between systems while engaging key biomechanical mechanisms associated with football-related lower-limb injuries [6,23,25].

All trials were started with a five-minute warm-up consisting of walking while wearing socks. Participants performed all tasks without shoes, wearing DAid Smart Socks only. This decision was made to ensure direct and stable sensor-to-surface contact and to avoid signal interference introduced by variations in football shoe structure, sole stiffness, and stud configuration. Testing without shoes allowed precise synchronization and comparison with the force platform by eliminating footwear-induced differences in plantar pressure transmission. Although this approach provides clearer information on sensor validity, it does not replicate real in-game footwear conditions.

For the single-leg squat (Figure 1a), each participant completed 10 repetitions on the right leg followed by 10 repetitions on the left leg, with 30 s rest between sets. For the squat with heel raise (Figure 1b), each participant performed the exercise continuously for 30 s. In the beginning of each set, the participant lifted right leg for 3 s, then lifted right leg for 3 s to provide the reference mark for sock sensor calibration and waveform synchronization. After completing the first trial of both exercises, the DAid^®^ Smart Socks were removed and re-applied before repeating the same exercise sequence to tackle the putting-on-induced repositioning of the socks. A metronome set to 60 beats per minute was used to standardize movement tempo.

The exercises were deliberately performed in a pre-defined manner, without randomisation, to closely mirror how FIFA 11+–style injury-prevention tasks are implemented in practice [23].

### 2.4. Instrumentation

Plantar pressure was measured using DAid^®^ Smart Socks (DAid^®^, Latvia), a fully textile-based wearable sensor system designed to measure plantar pressure distribution and foot loading characteristics during functional and sport-specific tasks. Each sock integrates six knitted pressure sensors positioned across the plantar surface—two under the heel, two under the midfoot (arch), and two under the metatarsal region—allowing detection of plantar pressure load shifts associated with for, e.g., foot pronation and supination. The pressure sensors were knitted using electroconductive WX-60 yarn (linear resistivity approximately 100 kΩ/m), connected by conductive pathways, made with electroconductive Shieldex 117/17 yarn (linear resistivity approximately 1 kΩ/m). The electroconductive yarns in the pressure sensors are integrated as rows of loops that are interconnected and form electric pathways, and the amount and quality of connections between the loops determine the electrical conductivity. The pressure applied to the sensors compresses the loops together, thus improving the connection between them, consequently the electrical conductance of these sensors increases. The typical sensitivity of the sensors varied within 0.5–3.0 μS/kPa but strongly depends on how the sock is strained during putting-on. The sensors and the conductive pathways are incorporated in the socks during the manufacturing process by a commercial sock knitting machine. Further technical details of the DAid^®^ Smart Socks system are described elsewhere [15,16,18]. The conductive textile pathways connect sensors to a compact data acquisition unit (6.6 × 4.0 × 1.3 cm) that attaches to the shoe via snap fasteners. The unit records electrical resistances of the sensors (in kOhms) with sampling rate up to 200 Hz per channel and transmits data wirelessly via Bluetooth to a proprietary software environment developed in LabVIEW (National Instruments, Austin, TX, USA) for synchronized signal acquisition and processing [16,21]. Further technical details of the DAid^®^ Smart Socks system are comprehensively described in Januskevica et al. (2020) [21].

The placement of the sensors on the plantar surface was visually inspected and manually adjusted to ensure that the forefoot sensors were placed beneath the metatarsal heads and the heel sensor along the midline of the calcaneus. This procedure was performed for all participants every time the socks were put on.

As the reference system, a 1.5 m entry-level force platform (RSscan International, Olen, Belgium, data acquisition frequency: 125 Hz, and pressure range: 0–200 N/cm^2^) was used, with a calibration factor of 3.2. Center of Pressure (CoP) data were recorded using Footscan software version 9.10.4, a repeatable tool for the assessment of plantar pressure distribution, and the normal values of the foot loading parameters [28]. A metronome application (Metronome Beats, Stonekick^®^, London, UK) controlled movement tempo at 60 beats per minute during all exercise trials. Synchronization between the two measurement systems was achieved through a standardized calibration procedure involving weight shifts and foot taps on the platform prior to each test.

As a result, the waveforms of the conductance of the six DAid^®^ Smart Socks sensors were recorded, along with the waveforms of X and Y coordinates of the CoP, provided by the force platform.

### 2.5. Calculation of the Center of Pressure (CoP) Waveform from Sock Data: Geometry

The position of the Center of Pressure (CoP) was computed from the six textile sensors integrated into the DAid^®^ Smart Socks, using the weighted vector summation method and CoP calculation approach described in Januskevica et al. and Semjonova et al. [21,22].

The positions of the six sensors were defined in a local coordinate system fixed to the sock sole. Vectors were assigned relative coordinates (mm) that represent the spatial distribution of sensors rather than their absolute locations on the anatomical foot. The relative positions were: e0 = (50, 100), e1 = (−50, 100), e2 = (50, 0), e3 = (−50, 0), e4 = (25, −100), e5 = (−25, −100), where the X-axis indicates the medial-lateral direction and the Y-axis indicates the anterior–posterior direction (Figure 2).

The CoP vector (in millimeters) was calculated as a weighted sum of the sensor direction vectors $e_{i}$ and sensor electrical conductance *U_i_*, that is proportional to the sensor applied pressure, according to the following equation:(1)$$\vec{Cop}=\frac{\sum U_{i}\vec{e_{i}}}{\sum U_{i}}$$
where $\vec{Cop}$—resulting CoP vector expressed in millimeters, $U_{i}$—conductance of the sensor *i*, proportional to the plantar pressure, in millisiemens, $e_{i}$—vector pointing from the coordinate origin to the *i*-th sensor position (*i* = 0–5).

Since the sock sensors are knitted into a flexible textile structure, the defined coordinates are arbitrary and expressed in relative units, representing the geometric relationship between sensors rather than the absolute spatial scale of the foot. The resulting CoP describes the relative movement and direction of the pressure center during dynamic or functional tasks, rather than its precise location on the plantar surface.

By this way, the waveforms of the X and Y coordinates of CoP were obtained form socks data for each participant, for each exercise, and each repetition.

### 2.6. Data Analysis Methods

To enable comparison with the reference force platform data, the platform and sock CoP signals were temporally aligned based on corresponding waveform features (e.g., peak timing and amplitude) to ensure synchronization between systems.

After synchronization, the waveforms were visually inspected to assess the location and spread of the CoP measured by both the smart socks and the force platform. In parallel, the average CoP coordinates in the X and Y directions, together with their corresponding standard deviations, were calculated for each waveform. Figure 3 illustrates an example of the variability in the average CoP X-coordinate obtained during a single right-leg squat.

Both the visual inspection and the analysis of CoP variability using diagrams similar to Figure 3 for other exercise types revealed substantial variation in the mean CoP location for both the socks and the platform. The variation in the mean CoP reported by the platform occurred because its outputs absolute CoP coordinates relative to a fixed global origin; therefore, the reported CoP values depended on the participant’s foot placement on the force plate. In contrast, variations in the CoP reported by the socks were attributed to differences in sensor placement between measurement sessions and to anatomical variation between participants. To compensate for these effects, each waveform was normalized individually by subtracting its own mean CoP value, computed over the full duration of that waveform.

The agreement between CoP values, measured using the DAid^®^ Smart Socks and by the force platform, was estimated using a variety of techniques:

*a. Time-series plots* were generated for each recording corresponding to a unique combination of participant, exercise type, and exercise repetition, for both the X and Y CoP coordinates. Each plot included the time-series data obtained from the DAid^®^ Smart Socks and from the force platform.

*b. Correlation diagrams* comparing the CoP coordinates derived from the DAid^®^ Smart Socks and the force platform were created for each combination of participant, exercise type, and exercise repetition.

*c. Bland–Altman plots* were generated for each waveform pair (sock–platform) to assess the agreement between the CoP measurements obtained from the smart socks and the force platform. For each time point, the difference between the two measurements (socks–platform) was plotted against their mean value, following the conventional Bland–Altman representation.

The mean difference (bias) and the standard deviation of the differences were calculated for each waveform pair. Using confidence level *α* = 0.95, the limits of agreement were defined as(2)$$LoA=\langle d\rangle \pm 1.96\;S\left(d\right),$$where *<d>* is the mean difference and *S(d)* is the standard deviation of the differences.

*d. The root mean square error (RMSE)* between the CoP values derived from the DAid^®^ Smart Socks and the force platform was calculated for each combination of participant, exercise type, and exercise repetition. The RMSE was computed as(3)$$RMSE=\sqrt{\frac{1}{N}\sum_{i=1}^{N}{\left({CoP}_{\mathrm{ss},i}-{CoP}_{p,i}\right)}^{2},}$$were indexes s and p denoting CoP data obtained using socks and platform, correspondingly. The error was reported both as an absolute RMSE and as a relative RMSE. As the relative error could not be calculated using the mean CoP value—because the normalization procedure centered each waveform such that its mean value was equal to zero—the relative RMSE was expressed as a percentage of the interquartile range Δq calculated as difference between 97.5% and 2.5% quantiles of the corresponding CoP waveform.

*e. Lin’s Concordance Correlation Coefficient (CCC)* [29,30] was used as a quantitative metric to numerically assess the agreement between the CoP measurements obtained from the DAid^®^ Smart Socks and the force platform. For each waveform pair (sock–platform), the CCC was calculated using the normalized CoP values in the X and Y directions. The CCC for each waveform pair was computed using the technique described in [30]. The coefficient CCC was calculated as follows:(4)$$CCC=\rho \cdot \chi_{\alpha }$$where *ρ* is the Pearson correlation coefficient between sock and platform CoP waveform and *χ_α_* is the accuracy coefficient is calculated using location shift *v* and scale shift *ω*:(5)$$\chi_{\alpha }=\frac{2}{v^{2}+\omega +\frac{1}{\omega }},\;v^{2}=\frac{{\left(\mu_{s}-\mu_{p}\right)}^{2}}{s_{s}s_{p}},\;\omega =\frac{s_{s}}{s_{p}}$$Here *μ_s_*, *s_s_*, *μ_p_*, and *s_p_* are the average CoP values and standard deviations for sock and platform waveforms, correspondingly. The confidence intervals for CCC were constructed at the level of significance *α* = 0.05 using Fisher’s transformation [29,30].(6)$$\lambda ={tanh}^{-1}\left(CCC\right),$$where *λ* is distributed normally with mean *μ_λ_* = *atahn*^−1^(*μ_CCC_*) and variance(7)$$\sigma_{\lambda }^{2}=\frac{1}{n-2}\left\{\frac{(1-\rho^{2}){CCC}^{2}}{(1-{CCC}^{2})\rho^{2}}+\frac{2{CCC}^{3}(1-CCC)v^{2}}{\rho {\left(1-{CCC}^{2}\right)}^{2}}-\frac{{CCC}^{4}v^{4}}{2\rho^{2}{\left(1-{CCC}^{2}\right)}^{2}}\right\}$$

The CCC values and corresponding confidence intervals were obtained for all participant–exercise–repetition combinations were used to compare the overall performance of the textile-based system relative to the force platform.

The interpretation of CCC values is still debated in the literature. In the present study, two complementary approaches were used. The strict criteria proposed in McBride 2005 [31] were applied to assess whether the DAid^®^ Smart Socks could serve as a replacement for the force platform in measuring the absolute CoP position. In parallel, a less stringent, correlation-oriented interpretation [27] was used to evaluate the relative correspondence between the CoP waveforms from the socks and the platform, reflecting their potential use for tracking changes in CoP position rather than absolute values. The corresponding CCC intervals are summarized in Table 1.

## 3. Results

### 3.1. The Dataset

In total, 160 waveforms were collected for each measurement modality, the DAid^®^ Smart Socks and the force platform. Each waveform was annotated with the participant identifier, exercise type, exercise repetition, leg (left or right), and whether it represented the CoP X or Y coordinate. The number of individual data points per waveform ranged from 1146 to 2104.

For the analysis, each waveform pair (socks–platform) was used to compute one set of metrics: RMSE and CCC. These metrics were then analyzed separately for the CoP X and CoP Y coordinates.

### 3.2. Time-Series Analysis of Single Squat CoP Waveforms—Left and Right Foot Comparison

During single-squat exercises performed with the left foot, the comparison of Center of Pressure waveforms in the mediolateral (CoP X) and anteroposterior (CoP Y) directions, obtained from the DAid^®^ Smart Socks and the force platform, is illustrated in Figure 4.

During single-squat exercises performed with the right foot, the comparison of Center of pressure waveforms in the mediolateral (CoP X) and anteroposterior (CoP Y) directions, obtained from the DAid^®^ Smart Socks and the force platform, is illustrated in Figure 5.

Both systems demonstrated a similar temporal pattern of CoP displacement in the mediolateral (X) and anteroposterior (Y) directions. The waveforms showed synchronized peaks and troughs across the entire trial duration, indicating consistent detection of load transfer and balance adjustments throughout the squat cycle. The overlay of CoP trajectories in the X–Y plane illustrates a close spatial correspondence between the two systems. The platform data exhibited slightly larger amplitude variations across both mediolateral and anteroposterior directions. The trajectory patterns and directions of CoP movement were closely matched between the two systems. Both the left and right foot analyses revealed comparable CoP waveform patterns between the DAid^®^ Smart Socks and the force platform. Figure 6 illustrates typical trajectories of CoP, indicating the fluctuation of CoP positions during single-leg squat exercises ranged approximately from –20 mm to 20 mm in the mediolateral (X) direction for each foot.

In the anteroposterior (Y) direction, visual comparison of CoP trajectories shows that, for right-leg–dominant participants, the left foot trajectories were more compactly clustered around the center, with fluctuations below ±40 mm along the Y-axis, and similar waveform patterns were observed for both measurement systems. For the right foot, the CoP spread was slightly more elongated in the anteroposterior direction (up to and over ±40 mm), while the overall directional pattern of CoP movement was similar for both measurement systems. The platform data exhibited slightly higher amplitude variation compared to the Smart Socks. Despite these minor differences, the spatial distribution and trajectory patterns remained consistent across both feet.

### 3.3. Time-Series Analysis of Squat CoP Motion—Both Feet

The Center of Pressure (CoP) trajectories recorded simultaneously from the DAid^®^ Smart Socks (red) and the force platform (blue) during squat exercises for both the left and right foot. The CoP excursions followed similar spatial patterns in both systems, with the variation in CoP position distributed predominantly along the anteroposterior (Y) axis and smaller mediolateral (X) deviations (Figure 7 and Figure 8).

For the left foot, the CoP displacement ranged approximately from −50 mm to +60 mm in the Y-direction and from −20 mm to +20 mm in the X-direction (Figure 9). For the right foot, the Y-direction excursions extended by the same range from −50 mm to +60 mm, as for the left foot, but with higher mediolateral variations within ±35 mm. The DAid^®^ Smart Socks closely reproduced the trajectory shape and direction measured by the force platform, showing consistent path overlap and similar amplitude ranges. Although the platform data exhibited slightly greater dispersion, particularly in the superior and inferior regions of the CoP path, the CoP trajectories from both systems were visually closely aligned (Figure 9).

### 3.4. Root Mean Square Error (RMSE) Between Smart Socks and Force Platform

Table 2 and Table 3 summarize the root mean square error, calculated using Equation (3) for each waveform pair “socks–platform”. The table represents RMSE in absolute form, expressed in mm, and in relative form, calculated as RMSE divided by interquartile range Δq = q_95_ − q_05_ for the corresponding waveform.

For mediolateral CoP (X-axis), the absolute RMSE values varied from 2.1 mm to 18.9 mm, comprising 8% to 46% of the typical span of CoP X waveform, measured using a force platform. For one waveform (participant Nr 2, single left leg squat), the relative RMSE reached 97%. The detailed analysis of this waveform showed a rapid change in CoP X position, measured using the platform, because of the change in the participant’s position.

For anteroposterior CoP (Y-axis), the absolute RMSE values varied from 4.3 mm to 34.2 mm, comprising 5% to 49% of the typical span of CoP Y waveform, measured using a force platform. For both the X and Y components, relatively large RMSE values were observed between the CoP data obtained from the smart socks and the force platform. Figure 10 summarizes the variation in both absolute and relative RMSE over all studied waveforms.

### 3.5. Concordance Between Smart Socks and Force Platform

A total of 160 recordings were analyzed to determine the correspondence between CoP measurements obtained from the DAid^®^ Smart Socks and the reference force platform (Table 4 and Table 5).

Across all test conditions and participants, the concordance correlation coefficient (CCC) ranged from 0.02 to 0.98. Out of all recordings, only 25 cases (16%) demonstrated acceptable agreement (moderate to very strong concordance, defined as CCC ≥ 0.90). For mediolateral CoP (X-axis), CCC values ranged from 0.04 to 0.96, with only 8 cases (10%) meeting this acceptability criterion, while for anteroposterior CoP (Y-axis), CCC values ranged from 0.02 to 0.99, with 17 cases (21%) showing acceptable concordance.

When applying less stringent criteria and interpreting CCC primarily as a measure of correlation strength, the pattern appears less pessimistic. For most waveforms—127 out of 160 (79%)—CCC values ≥ 0.6 indicated moderate or high correlation between the sock-based and platform-based CoP measurements, whereas only 10 waveforms (6%) showed poor correlation (CCC < 0.4). For the mediolateral CoP (X-axis), moderate or high correlation was observed in 58 out of 80 cases (73%), with 7 cases (9%) classified as poor. For the anteroposterior CoP (Y-axis), the corresponding numbers were 69 cases (86%) with moderate or high correlation and only 3 cases (4%) with poor correlation.

### 3.6. Scattering Diagrams and Bland–Altman Analysis of Concordance Between Smart Socks and Force Platform

The constructed scatter diagrams (“CoP by socks vs. CoP by platform”) allowed for a visual assessment of the relationship between the two measurement modalities. Figure 11 presents examples of the scatter diagrams for both CoP X and CoP Y. These examples correspond to exercises in which the CoP X measurements exhibited the best concordance (CCC = 0.96), median concordance (CCC = 0.73), and the worst concordance (CCC = 0.04).

For these same recordings, the corresponding CCC values for CoP Y were 0.86, 0.90, and 0.57, respectively.

The scatter plots reveal several notable features. Even in cases with high concordance, a degree of non-linearity between the CoP values obtained from the socks and those from the platform can be observed. In some plots, a systematic deflection from the identity line indicates a potential scaling discrepancy between the two measurement modalities. Other diagrams show evidence of saturation in the sock-derived CoP X values at larger deviations, suggesting limitations in the dynamic range of the textile sensors. Scatter plots corresponding to lower concordance values exhibited multiple clusters of points rather than a single coherent trend. In the present example, two distinct clusters correspond to two different foot positions adopted by the participant on the platform, each associated with its own mean CoP X level. The corresponding CoP waveform is shown in Figure 11.

Figure 12 presents the Bland–Altman plots for the same set of waveforms shown in Figure 11. The red lines indicate the limits of agreement, computed from the standard deviation of the Bland–Altman differences at a confidence level of 0.05 (see Equation (2)). Across all plots, the proportion of data points falling outside the limits of agreement ranged from 0.2% to 8%.

Despite the relatively small number of out-of-range points, the width of the limits of agreement is substantial when compared with the overall amplitude of the CoP signals. In the studied dataset, the span between the upper and lower limits typically reached approximately 25% of the total range of CoP variation during the exercises. This large spread demonstrates considerable disagreement between the values obtained from the socks and those obtained from the force platform.

Such wide limits of agreement are consistent with the low CCC values observed for these recordings and indicate that the textile-based system is not well suited to accurate measurement of absolute CoP position.

Figure 13 shows representative waveforms corresponding to one of the lowest concordance correlation coefficients observed in the dataset (CCC = 0.04). The waveforms illustrate a sudden shift in the CoP baseline measured by the force platform, caused by an abrupt lateral step performed by the participant. This change in foot position creates two distinct CoP regions, producing a strong bias between the platform and sock measurements and thereby substantially reducing the CCC.

It is noteworthy that when only the CoP variation around the baseline is considered (excluding the shift in absolute position) the concordance improves considerably, with CCC increasing to approximately 0.85. This indicates that the socks are able to capture relative changes in CoP dynamics reasonably well, even though they may fail to reproduce abrupt absolute displacements caused by changes in foot placement.

## 4. Discussion

This study aimed to evaluate the validity of the DAid^®^ Smart Socks in measuring plantar center of pressure during functional FIFA 11+ Part 2 exercises by comparing their outputs with a gold-standard force platform. The findings show that the Smart Socks consistently reproduced the relative CoP movement patterns detected by the force platform, demonstrating moderate-to-high correspondence for dynamic trajectory changes across most recordings. At the same time, the system exhibited limited accuracy in absolute CoP positioning, with large variability in concordance and error values, indicating that it cannot yet replace laboratory instruments for precise spatial CoP measurement.

These findings are in agreement with previous research validating the DAid^®^ smart textile systems across various applications [15,16,17,18]. Semjonova et al. [18] confirmed the reliability and validity of the DAid^®^ Smart Shirt for assessing shoulder girdle elevation compared with an optical motion capture system, demonstrating that textile-integrated strain sensors can accurately capture upper-limb kinematics. Similarly, Januskevica et al. [21] reported that the DAid^®^ Pressure Sock System (DPSS) effectively detected medial–lateral shifts in CoP during single-leg squat tests in female athletes, correlating well with physiotherapists’ assessments of pronation and foot control. More recently, Semjonova et al. [22] conducted a formative usability evaluation of the smart textile sock system, emphasizing its applicability for real-time self-correction during functional tasks and highlighting good user satisfaction. These studies and the present findings support the reliability, validity, and user-centered design of the DAid^®^ platform for biomechanical assessment in both clinical and athletic contexts.

From a sports medicine perspective, continuous monitoring of plantar loading and CoP dynamics provides valuable insights into movement control and injury prevention [32,33,34]. Altered CoP trajectories—particularly excessive medial displacement—are associated with overpronation and increased stress on the lower limb kinetic chain, contributing to conditions such as patellofemoral pain, medial tibial stress syndrome, and anterior cruciate ligament injuries [34]. By enabling accurate and portable CoP measurement, DAid^®^ Smart Socks could serve as a field-based alternative to laboratory systems, facilitating early detection of abnormal loading patterns and supporting individualized interventions. Such functionality aligns with the broader movement in sports technology toward wearable, real-time feedback systems for athlete monitoring and rehabilitation [29,30,31,32,33,34,35,36].

Despite the overall high agreement between the smart socks and force platform, indicating stable balance control and close correspondence between the two systems, minor discrepancies were observed, likely attributable to sensor noise, fabric deformation, or inconsistent skin–sensor contact. Similar technical limitations have been described in prior studies using textile-based sensors [37,38], suggesting that further optimization of calibration procedures, signal filtering, and sensor placement could enhance precision. In this context, the relatively large RMSE values observed for both mediolateral and anteroposterior CoP components reflect limited accuracy in absolute CoP alignment between the smart socks and the force platform, rather than an inability to capture relative CoP dynamics. The observed discrepancies between systems were especially pronounced during trials in which foot position shifted abruptly, leading to baseline offsets that markedly reduced CCC values. This effect was also visually evident in scatter plots associated with lower CCC values, which exhibited multiple clusters of points rather than a single coherent trend, reflecting changes in baseline CoP position rather than inconsistencies in relative CoP variation.

This highlights an important methodological constraint; the textile sensor array—fixed to the sock rather than the ground—cannot detect foot repositioning on the platform, whereas the force plate treats such changes as true CoP displacement due to higher sampling precision and surface sensitivity. When analysis focused solely on within-position CoP variation, excluding these baseline shifts, concordance improved substantially (CCC ≈ 0.85), reinforcing that the DAid^®^ Smart Socks reliably detect dynamic changes, but not absolute spatial repositioning of the foot. This distinction is critical for interpreting the socks’ performance in functional tasks that involve small balance adjustments rather than large relocations of the base of support [38].

Another limitation is that the two exercises included in this study—the single-leg squat and the squat with heel raise—represent controlled neuromuscular tasks and do not capture the high-speed, multidirectional, and agility-dependent demands of real football performance. In match and training contexts, players frequently perform rapid cutting, hopping, deceleration, and change-of-direction movements that impose substantially different foot-loading patterns and CoP trajectories. A key limitation of the study is that participants performed the exercises without football shoes. Football footwear can alter plantar pressure distribution and CoP patterns due to variations in stud design, midsole stiffness, and upper material. Therefore, the present findings cannot be directly generalized to conditions in which athletes wear football boots.

Future studies should validate the Smart Socks under different footwear configurations to assess whether measurement accuracy is maintained during football-specific on-field conditions—incorporate more dynamic and field-relevant assessments, such as the Side Hop Test, Figure-of-Eight Hop Test, or cutting agility drills, to evaluate whether DAid^®^ Smart Socks maintain accuracy under high-intensity, sport-specific loading conditions. Also, future research should involve larger and more diverse populations, athletes of varying performance levels, and a broader range of dynamic sport-specific movements, including running and change-of-direction tasks. Long-term monitoring should also be incorporated to determine whether changes in plantar pressure or CoP patterns can predict injury risk, reflect recovery progress, and reveal relationships with movement asymmetries, pronation–supination (over pronation and over supination) control, and postural stability during performance and rehabilitation.

## 5. Conclusions

This study evaluated the preliminary validity of the DAid^®^ Smart Socks by quantifying their agreement with a gold-standard force platform during functional FIFA 11+ injury-prevention tasks in youth football players. The results showed that the Smart Socks consistently reproduced the directional and temporal characteristics of CoP trajectories, demonstrating moderate-to-high correspondence with the force platform across most recordings. At the same time, the level of agreement varied notably between individuals, reflecting the influence of foot morphology and sensor positioning inherent to the six-sensor textile design. Overall, the findings indicate that while the system is not suitable for precise estimation of the absolute spatial location of the CoP on the plantar surface, due to its fixed sensor geometry and the use of waveform normalization, it consistently captures relative CoP dynamics, such as the direction, timing, and magnitude of CoP shifts during movement. This supports its applicability in contexts where monitoring movement patterns and load-shift tendencies over time is of primary interest, rather than exact CoP coordinates.

## Figures and Tables

**Figure 1 healthcare-14-00076-f001:**
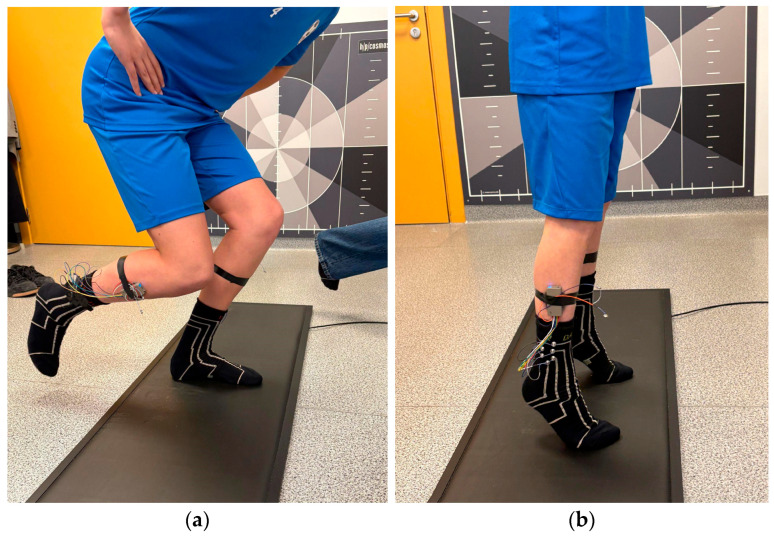
FIFA 11+ Part 2 exercises: (**a**) single-leg squat and (**b**) squat with heel raise.

**Figure 2 healthcare-14-00076-f002:**
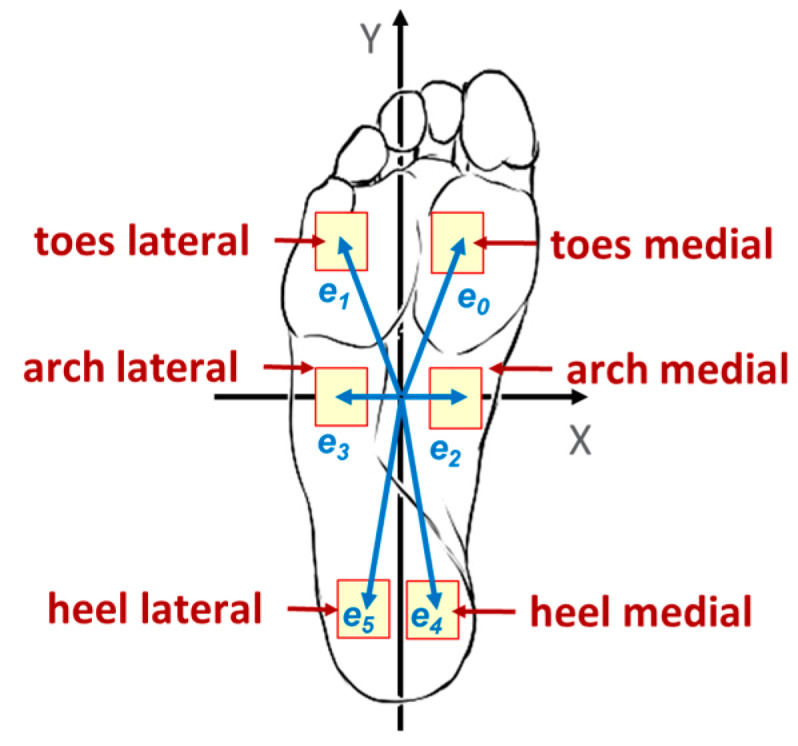
Foot-related coordinate system and designation of sensor vectors.

**Figure 3 healthcare-14-00076-f003:**
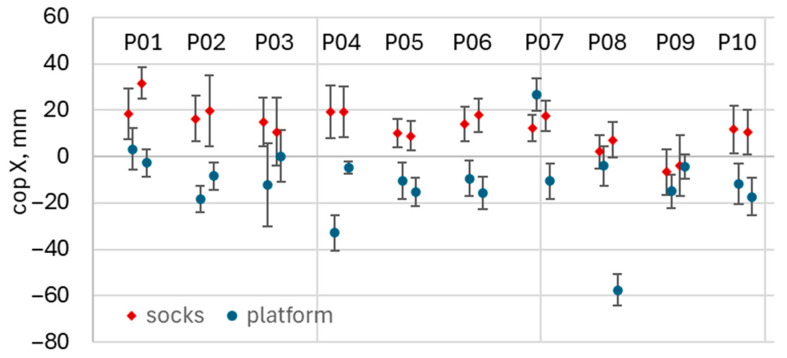
The mean X-coordinate of the CoP in a single right-leg squat series for different participants, denoted P01–P10. Each participant performed squat series two times. Red and blue dots represent recording series, obtained using socks and platform, correspondingly. Note high variability of the mean CoP X coordinate for platform data.

**Figure 4 healthcare-14-00076-f004:**
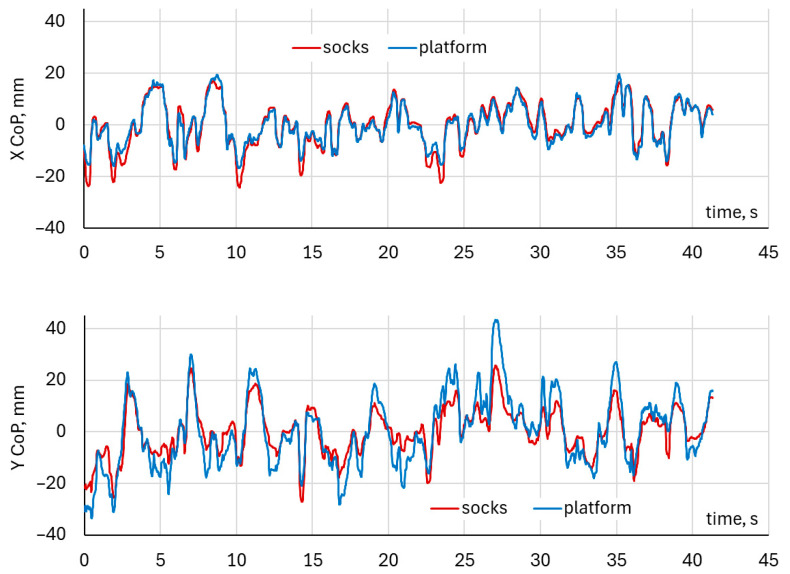
Example of CoP waveforms for left foot in a single-leg squat exercise.

**Figure 5 healthcare-14-00076-f005:**
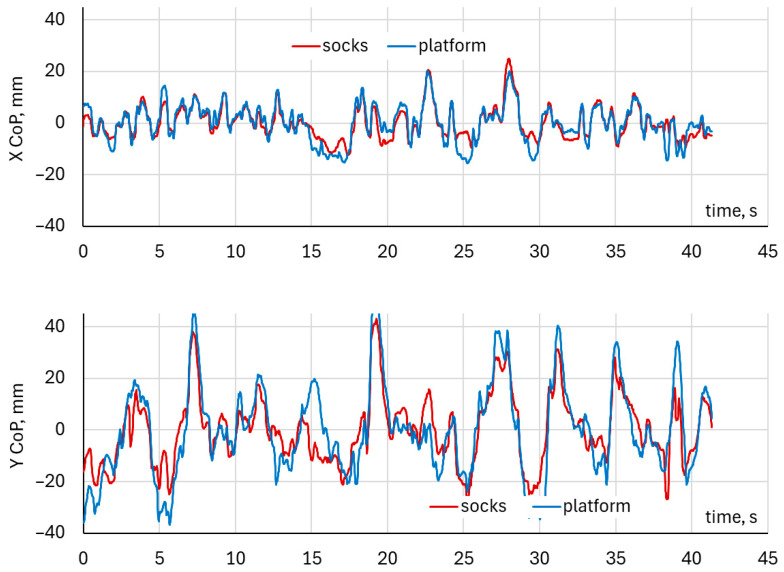
Example of CoP waveforms for the right foot in a single-leg squat exercise.

**Figure 6 healthcare-14-00076-f006:**
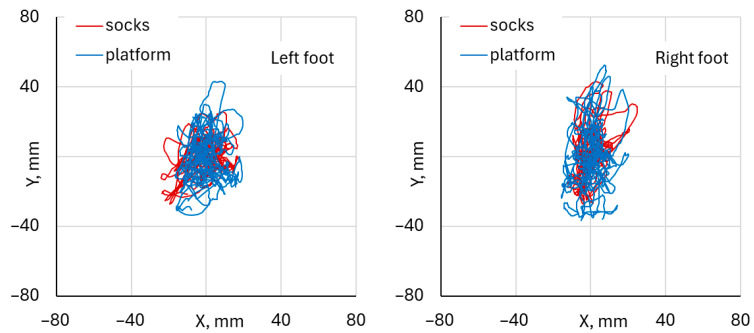
Example of the fluctuation of CoP position during single-leg squat exercise, performed by the same participant.

**Figure 7 healthcare-14-00076-f007:**
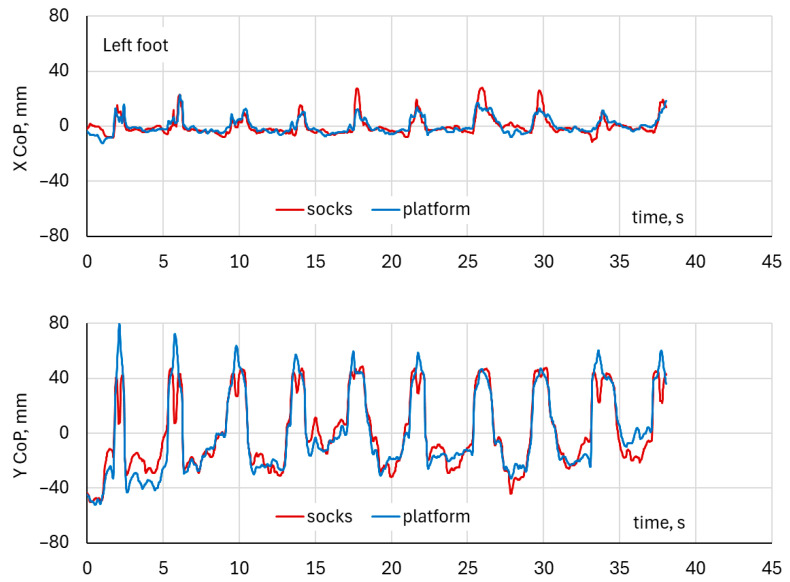
Example of CoP waveforms for the left foot in a two-leg squat exercise.

**Figure 8 healthcare-14-00076-f008:**
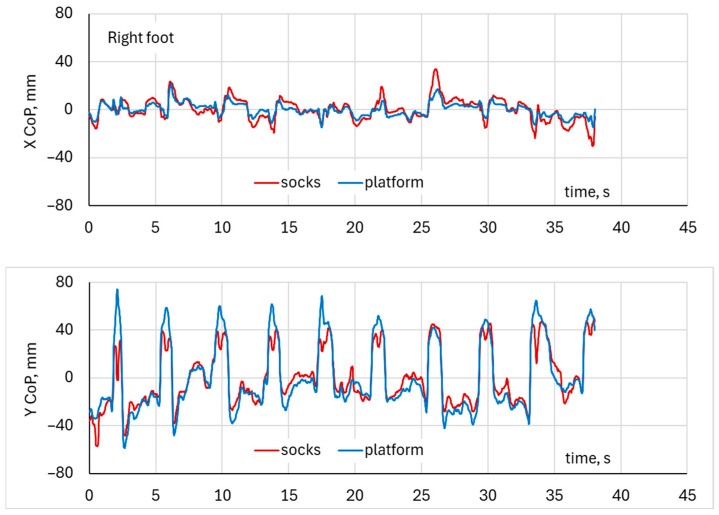
Example of CoP waveforms for the right foot in a two-leg squat exercise.

**Figure 9 healthcare-14-00076-f009:**
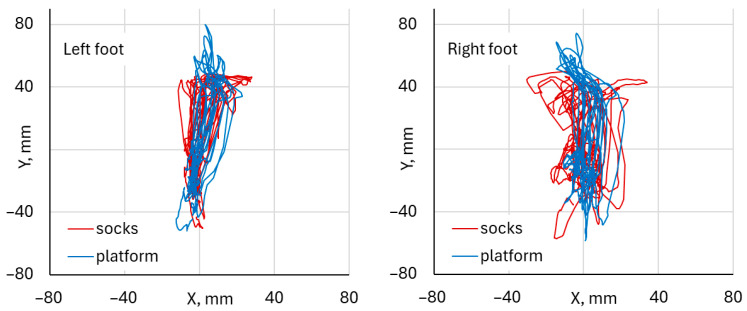
Example of the fluctuation of CoP position during both leg squat exercises.

**Figure 10 healthcare-14-00076-f010:**
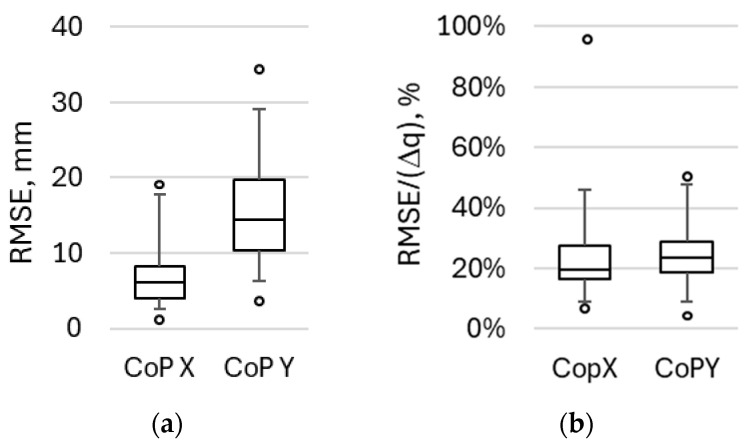
Box-and-Whisker plots for absolute (**a**) and relative (**b**) root mean square error between CoP measured using DAid^®^ Smart Socks and the reference force platform. Relative error was calculated using interquartile range Δq = q_97.5_ − q_2.5_. The whiskers correspond to a 97.5–2.5 interquartile range.

**Figure 11 healthcare-14-00076-f011:**
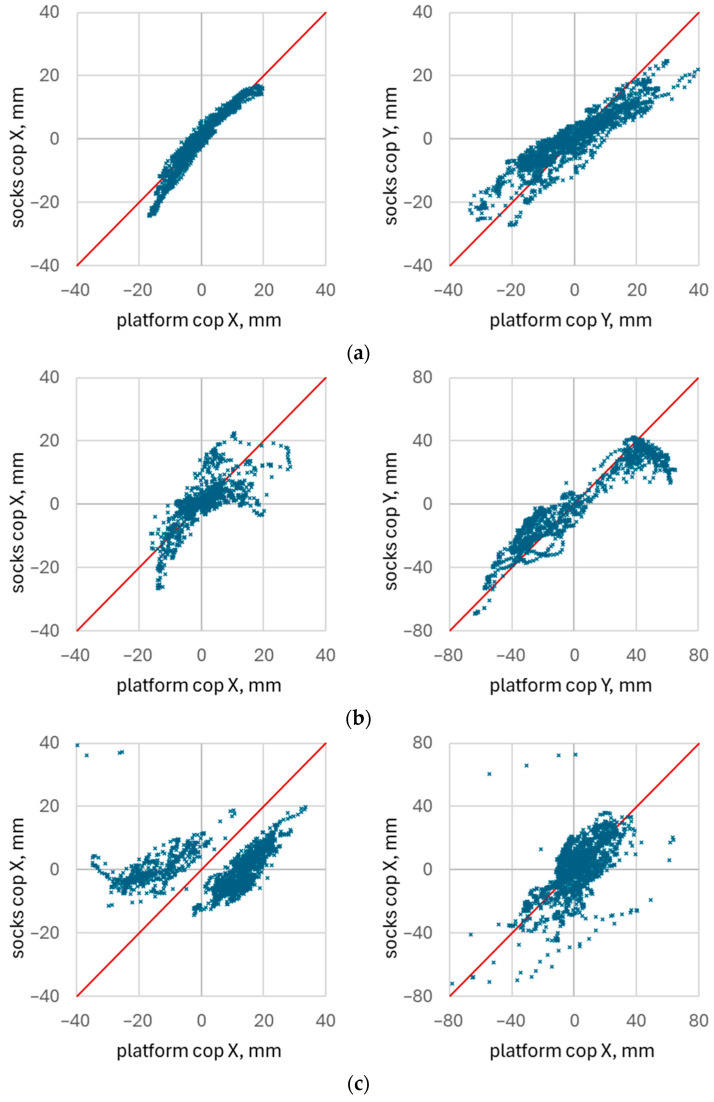
Scattering diagrams illustrate concordance between smart socks and force platform for the best (CCC = 0. 96, (**a**)), median (CCC = 0. 73, (**b**)), and worst (CCC = 0.04, (**c**)) concordance for CoP X. Concordance coefficients for CoP Y were 0.86 (**a**), 0.90 (**b**), and 0.57 (**c**).

**Figure 12 healthcare-14-00076-f012:**
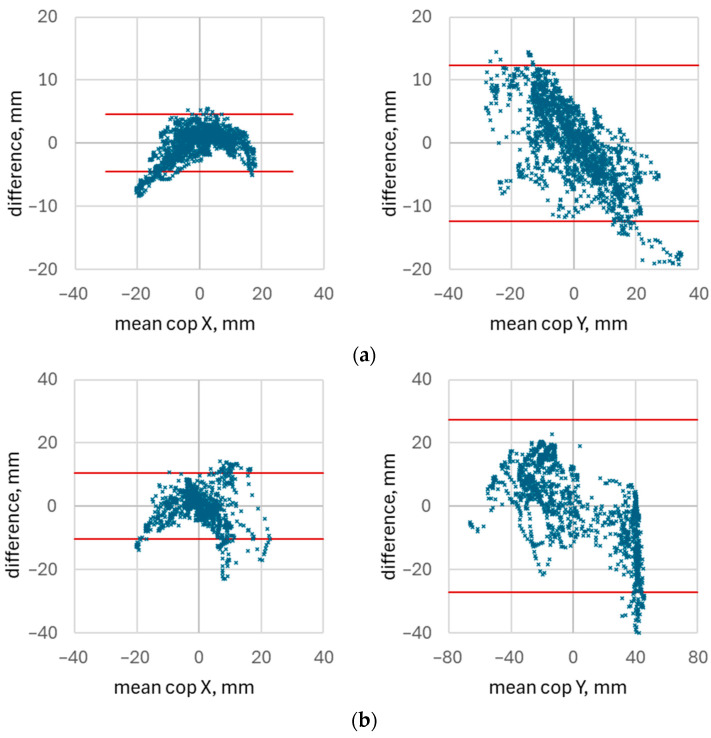
Bland—Altman plots illustrating agreement CoP data measured using smart socks and force platform for the best (CCC = 0. 96, (**a**)), median (CCC = 0.73, (**b**)), and worst (CCC = 0.04, (**c**)) concordance for CoP X. Concordance coefficients for CoP Y were 0.86 (**a**), 0.90 (**b**), and 0.57 (**c**).

**Figure 13 healthcare-14-00076-f013:**
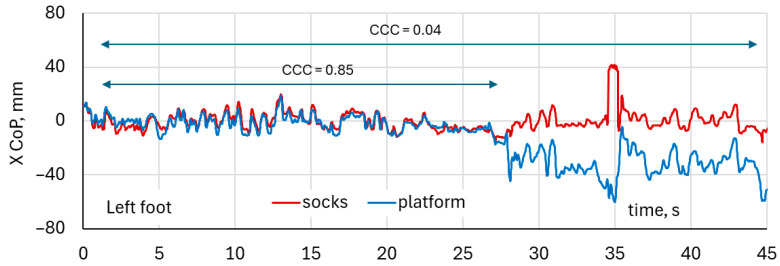
Example of poor agreement between CoP measured using smart socks and force platform due to a sudden change in the participant’s position (side step), detected by the platform as a bias in the average CoP X coordinate.

**Table 1 healthcare-14-00076-t001:** The CCC boundaries for classification of degree of concordance.

“Strict” Approach [28] Estimation of Concordance		“Loose” Approach, Adapted from [27] Estimation of Correlation	
0.99 and higher	Almost perfect	0.80 and higher	High
0.95 to 0.99	Substantial	0.60 to 0.80	Moderate
0.90 to 0.95	Moderate	0.40 to 0.60	Fair
less then 0.90	Poor	less then 0.30	Poor

**Table 2 healthcare-14-00076-t002:** RMSE estimation of the difference between CoP measured using DAid^®^ Smart Socks and the reference force platform. The values of RMSE are given in mm; the relative RMSE (in brackets) was calculated using interquartile range Δq = q_95_ − q_05_. CoP X—platform X.

Exercise	P01	P02	P03	P04	P05	P06	P07	P08	P09	P10
Right leg squat	3.5 (9%)	2.1 (10%)	17.8 (54%)	6.2 (20%)	5.8 (14%)	4.2 (12%)	9.4 (20%)	7.4 (18%)	4 (14%)	2.6 (10%)
	4.9 (25%)	3.1 (14%)	8.2 (18%)	5.9 (29%)	6.2 (16%)	3.2 (10%)	2.3 (8%)	7.7 (17%)	4.6 (16%)	2.5 (8%)
Left leg squat	11.1 (37%)	18.8 (97%)	11.6 (35%)	3.4 (13%)	5.9 (22%)	3.2 (15%)	4.1 (12%)	7.8 (27%)	6.9 (29%)	2.9 (13%)
	8.4 (18%)	3.5 (16%)	4.1 (13%)	3.7 (15%)	5.7 (22%)	2.8 (17%)	3.3 (18%)	10.5 (35%)	3.9 (17%)	8.1 (23%)
Both leg squat, right leg	6.9 (19%)	3.7 (17%)	11.4 (46%)	6.5 (19%)	6.3 (21%)	5.2 (31%)	17.7 (37%)	5.7 (15%)	7.2 (27%)	8.4 (25%)
	12.8 (27%)	4.2 (20%)	8.5 (28%)	3.9 (17%)	9 (23%)	4.1 (30%)	11.3 (28%)	7.5 (21%)	4.8 (17%)	8.3 (24%)
Both leg squat, left leg	15 (36%)	5.3 (20%)	6 (17%)	8.3 (21%)	7.4 (17%)	3 (19%)	5.6 (15%)	9.7 (23%)	5.1 (32%)	6.9 (21%)
	11.4 (31%)	6.9 (28%)	9 (37%)	9.6 (34%)	8.5 (18%)	3.5 (28%)	5.4 (15%)	8.2 (20%)	3.9 (18%)	6.7 (18%)

**Table 3 healthcare-14-00076-t003:** RMSE estimation of the difference between CoP measured using DAid^®^ Smart Socks and the reference force platform. The values of RMSE are given in mm; the relative RMSE (in brackets) was calculated using interquartile range Δq = q_95_ − q_05_. CoP Y—platform Y.

Exercise	P01	P02	P03	P04	P05	P06	P07	P08	P09	P10
Right leg squat	7.7 (19%)	8.7 (27%)	34.2 (41%)	13.8 (27%)	14.6 (26%)	10.6 (32%)	15.4 (39%)	10.5 (21%)	9.6 (18%)	7.1 (26%)
	19.7 (32%)	8.1 (22%)	18.3 (32%)	11.4 (48%)	10.9 (16%)	8.7 (45%)	6.3 (20%)	6.4 (14%)	15.8 (27%)	6.7 (24%)
Left leg squat	12.2 (27%)	17.2 (29%)	20.5 (30%)	9.3 (22%)	15.6 (25%)	9.1 (23%)	11.5 (24%)	14.8 (20%)	14.3 (23%)	8.1 (28%)
	12.3 (23%)	7.7 (15%)	10 (20%)	4.8 (12%)	11.2 (22%)	4.3 (14%)	8.8 (19%)	9.9 (13%)	8.6 (25%)	11.3 (19%)
Both leg squat, right leg	24 (20%)	10.8 (17%)	27.9 (49%)	14.3 (17%)	11.1 (8%)	22.4 (44%)	18 (26%)	14.5 (15%)	16.1 (22%)	23.7 (39%)
	29 (23%)	19.7 (29%)	28.5 (42%)	14.1 (16%)	11.5 (10%)	23.4 (49%)	25.3 (37%)	16.1 (15%)	11.1 (15%)	25.7 (38%)
Both leg squat, left leg	27.1 (23%)	13.9 (19%)	25.3 (24%)	16.9 (21%)	11.2 (9%)	18.3 (36%)	17.9 (23%)	15.4 (16%)	18.2 (24%)	24.9 (34%)
	26 (19%)	21 (24%)	34.2 (40%)	17.8 (25%)	6.6 (5%)	20.5 (39%)	19.5 (25%)	20 (22%)	11.3 (15%)	22.3 (24%)

**Table 4 healthcare-14-00076-t004:** Values and confidence intervals (*α* = 0.05) for concordance correlation coefficients (CCC). The colors indicate the groups with good and moderate concordance (CCC ≥ 0.9, green), and weak correlation (CCC < 0.4, orange. CoP X—platform X).

Exercise	P01	P02	P03	P04	P05	P06	P07	P08	P09	P10
Right leg squat	0.94	0.94	0.24	0.74	0.84	0.9	0.64	0.76	0.89	0.95
	0.93–0.94	0.93–0.94	0.2–0.27	0.71–0.76	0.83–0.85	0.89–0.9	0.63–0.66	0.75–0.78	0.88–0.9	0.95–0.96
	0.7	0.88	0.77	0.44	0.77	0.92	0.96	0.75	0.8	0.95
	0.68–0.72	0.86–0.89	0.75–0.79	0.43–0.46	0.76–0.78	0.92–0.93	0.96–0.96	0.74–0.76	0.78–0.81	0.95–0.96
Left leg squat	0.54	0.04	0.66	0.89	0.73	0.82	0.9	0.63	0.71	0.92
	0.51–0.58	0.01–0.07	0.64–0.68	0.88–0.89	0.71–0.75	0.81–0.83	0.89–0.91	0.6–0.66	0.68–0.73	0.91–0.92
	0.77	0.86	0.9	0.8	0.69	0.88	0.87	0.31	0.78	0.71
	0.76–0.78	0.85–0.87	0.89–0.91	0.79–0.8	0.67–0.71	0.87–0.89	0.86–0.88	0.26–0.35	0.77–0.79	0.69–0.73
Both leg squat, right leg	0.77	0.77	0.14	0.84	0.68	0.52	0.08	0.78	0.62	0.56
	0.75–0.79	0.75–0.79	0.09–0.19	0.82–0.85	0.66–0.7	0.48–0.56	0.03–0.13	0.76–0.79	0.59–0.65	0.53–0.59
	0.48	0.79	0.4	0.87	0.58	0.52	0.35	0.65	0.8	0.56
	0.44–0.52	0.77–0.8	0.36–0.44	0.85–0.88	0.56–0.6	0.48–0.56	0.31–0.39	0.62–0.67	0.79–0.81	0.53–0.59
Both leg squat, left leg	0.39	0.73	0.85	0.72	0.68	0.76	0.83	0.51	0.6	0.59
	0.35–0.43	0.71–0.76	0.84–0.87	0.7–0.74	0.67–0.69	0.74–0.79	0.82–0.85	0.48–0.54	0.57–0.63	0.56–0.63
	0.45	0.7	0.46	0.59	0.63	0.62	0.87	0.62	0.81	0.71
	0.4–0.49	0.67–0.72	0.42–0.5	0.55–0.62	0.61–0.65	0.59–0.65	0.86–0.88	0.6–0.65	0.79–0.82	0.69–0.73

**Table 5 healthcare-14-00076-t005:** Values and confidence intervals (α = 0.05) for concordance correlation coefficients (CCC). The colors indicate the groups with good and moderate concordance (CCC ≥ 0.9, green), and weak correlation (CCC < 0.4, orange. CoP Y—platform Y).

Exercise	P01	P02	P03	P04	P05	P06	P07	P08	P09	P10
Right leg squat	0.82	0.65	0.15	0.49	0.61	0.6	0.52	0.75	0.84	0.74
	0.8–0.83	0.62–0.67	0.11–0.19	0.45–0.53	0.58–0.64	0.57–0.63	0.49–0.54	0.73–0.77	0.82–0.85	0.72–0.76
	0.02	0.72	0.33	0.42	0.84	0.49	0.86	0.9	0.6	0.8
	−0.02–0.06	0.7–0.75	0.28–0.37	0.39–0.46	0.83–0.85	0.47–0.52	0.85–0.87	0.9–0.91	0.58–0.63	0.78–0.81
Left leg squat	0.63	0.57	0.54	0.8	0.6	0.81	0.78	0.78	0.65	0.8
	0.6–0.66	0.54–0.6	0.51–0.57	0.78–0.81	0.57–0.63	0.8–0.82	0.76–0.8	0.77–0.8	0.63–0.68	0.79–0.81
	0.56	0.87	0.73	0.93	0.73	0.9	0.84	0.91	0.76	0.87
	0.53–0.58	0.86–0.88	0.71–0.75	0.92–0.93	0.71–0.75	0.89–0.91	0.83–0.85	0.9–0.91	0.74–0.78	0.86–0.88
Both leg squat, right leg	0.88	0.93	0.64	0.91	0.98	0.62	0.85	0.93	0.82	0.7
	0.87–0.89	0.92–0.93	0.62–0.65	0.9–0.91	0.98–0.98	0.59–0.64	0.84–0.86	0.92–0.93	0.8–0.83	0.69–0.71
	0.84	0.8	0.65	0.92	0.97	0.67	0.76	0.91	0.91	0.7
	0.83–0.86	0.79–0.81	0.63–0.67	0.91–0.93	0.97–0.97	0.65–0.68	0.74–0.77	0.9–0.91	0.9–0.92	0.69–0.72
Both leg squat, left leg	0.84	0.9	0.81	0.87	0.98	0.73	0.85	0.92	0.79	0.75
	0.83–0.85	0.89–0.91	0.79–0.82	0.85–0.87	0.97–0.98	0.72–0.75	0.83–0.86	0.91–0.93	0.77–0.8	0.74–0.76
	0.88	0.84	0.47	0.83	0.99	0.72	0.84	0.86	0.92	0.83
	0.87–0.89	0.83–0.85	0.44–0.51	0.82–0.85	0.99–0.99	0.71–0.73	0.83–0.85	0.85–0.87	0.91–0.93	0.82–0.84

## Data Availability

The data related to the project “Smart textile solutions as biofeedback method for injury prevention for Latvian football youth league players”, project No. lzp-2023/1-0027 are available from the corresponding author upon request. Data management plan identification number: https://doi.org/10.5281/zenodo.11082395 (accessed on 4 February 2024).

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
