# Peer review of "Evaluation of DAid® Smart Socks for Foot Plantar Center of Pressure Measurements in Football-Specific Tasks: A Preliminary Validation Study"

_healthcare, 2025, doi:10.3390/healthcare14010076_

Round 1
Reviewer 1 Report
Comments and Suggestions for Authors
General comments
Overall, the manuscript is well structured. Descriptions are mostly clear but sometimes incomplete, lacking important information. Most importantly, the study has serious methodological flaws. Hence, the study should not be published in its current state. However, these flaws can be corrected, resulting in a major revision suggestion.
The introduction requires some refinements to strengthen the rationale for specifically investigating the selected movements. Moreover, the investigated movements are not football-specific. Instead, they are general movements considered relevant in football (-> part of FIFA 11+).
The methods have serious flaws. It is recommended that the authors consult the GRRAS guidelines for validation studies. These guidelines are not perfect, but they will help. Further recommendations are described below.
The suggested revisions will affect the results and subsequent sections to a degree that necessitates separate review of those sections after revision.
Specific Comments
Introduction
3rd paragraph: Gold standard is the best (only one: force plate measurements), the others are common practices but not gold standard. Therefore, define force plates as the gold standard and the others as alternative practices that also fail to overcome some of the same limitations.
6th paragraph: This paragraph is important and not detailed enough. Detailed information about the current knowledge of DAid® Smart Socks accuracy is missing. Does nothing exist in the literature? If nothing exists, then validation should focus on more general movements before investigating sports-specific movements. You say “remains limited”. Be more precise, what exactly exists? Elaborate in more detail.
7th paragraph: Like in the previous paragraph, you say “little research”, but you do not provide references nor what this little research found out. Again, what exactly is known? Elaborate in more detail. Moreover, this paragraph should provide information on why to focus on FIFA 11+ Part 2 and why specifically the selected two exercises are relevant.
Methods
It is not clear how the analysis handled the repeated values from each participant. Did you derive one value per participant to run a statistical test (then, n would be 10), or did you treat repeated values from each participant as individual data points in the analysis (then, n would be much more depending on the number of data points (repetitions) per participant). Since this is unclear whether your sample size is sufficient. In validation studies on technological equipment, you can handle repeated values from a single participant as independent, which would usually be a violation of statistical rules (you can find examples and explanations in the literature).
Participants: 1) A-priori power analysis is missing. 2) The sample size is not sufficient (if you calculate a single value per participate, which would be the norm, leading to n=10), leading to an underpowered study. 3) If n=10, sensitivity power analysis is highly recommended to know what effects are detectable at the common power level of 1-beta=0.80. 4) Approval from an institutional ethics review board is missing.
Intervention: 1) Describe the sequence following the actual time sequence, starting with the warm-up, then the exercises. 2) Why did you not randomize the sequence of the two exercises as well as legs during the first exercise? If you can collect more data (for example from another 10 participants), change the sequence of tasks and legs. Then you solve two problems: Missing sequence randomization and too small sample size.
Instrumentation: 1) Add the measurement/sensitivity range of the sensors used in the socks. 2) Clarify the sensor type. 3) Clarify how the socks are designed to make sure users are wearing the correctly (with the sensors at the right location). 4) Detailed information about the “entry-level force platform” is missing. Since it seems to be none of the commonly used systems that would be considered gold standard, you should provide technical details in addition to validation data of the platform itself (measurement accuracy).
Analysis: 1) The procedure (from raw data to final data point for analysis) is unclear. After measurement you have a data time series throughout the exercise across multiple repetitions. Each frame has a different fractal CoP value. How many values did you use for the analysis (the entire time series, one CoPx per participant, or one CoPx per repetition), and – if not the entire time series – how did you derive those values (e.g., the average across frames) or – if the entire time series – how did you combine time series data from multiple participants? Based on the results, I can only assume that you tested the entire time series for each participant separately. It must be clearly described what you did. 2) Pearson correlation is not a suitable test for the current type of validation study. Pearson tests linear relationship. You need to test exact agreement. You need to conduct tests like ICC (be careful about the specification because some test consistency instead of exact agreement) or CCC (tests always exact agreement). In addition, you should calculate descriptive information such as RMSE and Bland-Altman’s limits of agreement.
Other aspects in this chapter are irrelevant as Pearson’s correlation needs to be replaced by proper statistical tests. However, generally there are the following problems: 3) L197-198: The highest p value observed is a result and does not belong to the methods section. Just describe the significance level you used without results. 4) L198-199: r = 0.05 is NOT the critical value for non-significance. The correlation coefficient r expresses the effect size of the correlation, not its significance, which is expressed by p. 5) L199-201: Unclear what that means, but it seems to be a result or even an interpretation (-> discussion section), so just remove. 6) L201-205: Not necessary to explain the range of Pearson’s correlation coefficients (that is undergraduate-level knowledge, first statistics course). A reference for the interpretation benchmarks is missing.
Comments on the Quality of English LanguageWriting flaws indicate the need for revising grammar and style (e.g., missing main clause; L101-102).
Author Response
Thank you so much for your review! It took time to revise the manuscript accordingly, but that was worth it. Thank you!
Comment 1: The introduction requires some refinements to strengthen the rationale for specifically investigating the selected movements. Moreover, the investigated movements are not football-specific. Instead, they are general movements considered relevant in football (-> part of FIFA 11+).
Reply 1: We thank the reviewer for this valuable observation. To clarify the rationale for the selected movements, we expanded the introduction by explicitly explaining why the single-leg squat and squat with heel raise were chosen and how they relate to football injury-prevention contexts. We highlight that these tasks are part of FIFA 11+ Part 2 and are widely used in football to assess neuromuscular control and lower-limb stability, even though they are not strictly football-specific actions. A new paragraph has been added to the Method section (lines 172-184), strengthening the justification for the movement selection.
Comment 2: The methods have serious flaws. It is recommended that the authors consult the GRRAS guidelines for validation studies. These guidelines are not perfect, but they will help. Further recommendations are described below.
Reply 2:
We thank the reviewer for pointing us to the GRRAS guidelines. Following this recommendation, we consulted the original publication “Guidelines for Reporting Reliability and Agreement Studies (GRRAS)” (Kottner J., Audigé L., Brorson S., Donner A., Gajewski B.J., Hróbjartsson A., Roberts C., Shoukri M., Streiner D.L. (2011). Guidelines for Reporting Reliability and Agreement Studies (GRRAS). Journal of Clinical Epidemiology, 64(1), 96–106)
We revised our Methods and Results sections accordingly the structure, given in the guidelines.
We would like to note that our study does not aim to evaluate reliability (i.e., repeatability or reproducibility of measurements within the same instrument), but rather the level of agreement between two different measurement modalities. Therefore, we focused on statistical techniques appropriate for agreement analysis. We replaced the previously used Pearson correlation with the Concordance Correlation Coefficient (CCC), which directly quantifies both precision and accuracy relative to the line of identity and is therefore better suited for assessing agreement.
In addition, we implemented Bland–Altman analysis, including Bland–Altman plots with limits of agreement, to visualize systematic bias and the dispersion of differences across the measurement range. These adjustments align our analysis more closely with the GRRAS recommendations for agreement studies.
The suggested revisions will affect the results and subsequent sections to a degree that necessitates separate review of those sections after revision.
Specific Comments
Introduction
Comment 3: 3rd paragraph: Gold standard is the best (only one: force plate measurements), the others are common practices but not gold standard. Therefore, define force plates as the gold standard and the others as alternative practices that also fail to overcome some of the same limitations.
Reply 1: We thank the reviewer for this important clarification. The introduction has been revised to explicitly define force plates as the only gold-standard method for plantar pressure and CoP assessment. The paragraph has also been rewritten to describe instrumented walkways and in-shoe systems as alternative practices, acknowledging that they share similar limitations and therefore should not be classified as gold-standard tools. These revisions have been implemented in the Introduction, replacing lines 80–89.
Comment 4: 6th paragraph: This paragraph is important and not detailed enough. Detailed information about the current knowledge of DAid® Smart Socks accuracy is missing. Does nothing exist in literature? If nothing exists, then validation should focus on more general movements before investigating sports-specific movements. You say, “remains limited”. Be more precise, what exactly exists? Elaborate in more detail.
Reply 4: We thank the reviewer for highlighting the need to better contextualize existing evidence on DAid® Smart Socks performance. We expanded the 6th paragraph of the Introduction to provide a detailed overview of the current literature. This includes: performance evaluation of the DAid® Pressure Socks System (DPSS) during walking, demonstrating moderate-to-excellent ICC values (Eizentals et al., 2021; Gait & Posture) [15], DAid pressure socks system studies validating the system’s ability to detect medial–lateral CoP shifts in functional tasks (Januskevica et al., 2020) [20] and usability and feasibility evaluations relevant to functional performance (Semjonova et al., 2022) [21]. We clarified that existing studies primarily address controlled or low-complexity movements (e.g., walking) and highlighted why additional validation in sport and football-relevant functional tasks is needed. These enhancements were added after line 105 till 120 of the Introduction.
Comment 5: 7th paragraph: Like in the previous paragraph, you say “little research”, but you do not provide references nor what this little research found out. Again, what exactly is known? Elaborate in more detail. Moreover, this paragraph should provide information on why to focus on FIFA 11+ Part 2 and why specifically the selected two exercises are relevant.
Reply 5: Thank you for this helpful comment. We significantly expanded the Introduction to (1) clarify what previous research on DAid® Smart Socks has investigated and (2) justify the relevance of the selected FIFA 11+ Part 2 exercises. Specifically, we now describe existing studies showing that DAid® Smart Socks can detect medial–lateral CoP shifts (Eizentals et la.,. 2021and Januskevica et al., 2020) and support functional feedback (Semjonova et al., 2022), while emphasising the absence of validation during football-relevant neuromuscular-control tasks. We also added a detailed rationale for selecting the single-leg squat and squat with heel raise, highlighting their role in FIFA 11+ Part 2 and their importance for assessing balance, alignment, and foot-loading mechanics in youth football players.
Methods
Comment 6: It is not clear how the analysis handled the repeated values from each participant. Did you derive one value per participant to run a statistical test (then, n would be 10), or did you treat repeated values from each participant as individual data points in the analysis (then, n would be much more depending on the number of data points (repetitions) per participant). Since this is unclear whether your sample size is sufficient. In validation studies on technological equipment, you can handle repeated values from a single participant as independent, which would usually be a violation of statistical rules (you can find examples and explanations in the literature).
Reply 6: Thank you for this insightful comment. We have clarified the aim of the study in lines 141–145, and the overall design of the study in lines 150-162. As noted, our objective is to evaluate the performance of the device itself, rather than the performance of the athletes. The parameter of interest is the measure of agreement between the DAid® Smart Socks and the force platform (gold standard). Participant involvement was necessary only to introduce realistic variability and to assess how the agreement metric behaves under different conditions. Following your recommendation, we replaced the Pearson correlation coefficient with the Concordance Correlation Coefficient (CCC), which better reflects true agreement. The CCC was computed for each unique combination of Participant – Exercise – Repetition. This structure allowed us to examine how agreement varies as a function of (i) the individual participant, (ii) the specific exercise and leading-leg condition, and (iii) the putting-on-induced repositioning of the sock. Although each participant contributed multiple measurements, these measurements were not treated as simple repetitions of the same condition. Instead, each data point corresponded to a distinct combination of factors, and therefore represented a unique experimental condition. This approach is consistent with practices in preliminary validation studies of technological devices, where repeated observations from the same individual—when arising from different controlled conditions—can be treated as independent for the purpose of estimating device-related variability.
Comment 7: Participants: 1) A-priori power analysis is missing. 2) The sample size is not sufficient (if you calculate a single value per participate, which would be the norm, leading to n=10), leading to an underpowered study. 3) If n=10, sensitivity power analysis is highly recommended to know what effects are detectable at the common power level of 1-beta=0.80. 4) Approval from an institutional ethics review board is missing.
Reply 7: We appreciate the reviewer’s detailed remarks regarding sample size, power, and ethical approval. Concerning the power analysis, one would like to clarify that the present work was conceived and conducted as a preliminary validation study of the DAid® Smart Socks, primarily aimed at estimating agreement with a force platform and exploring sources of variability (participant, exercise, sock repositioning). The detailisation of the scope of the study is added to the method section (lines 150-161). At the time of data collection, no robust empirical information was available on the expected variability of the agreement measure (CCC) in this specific application, which made a conventional a-priori power calculation difficult. We have now clarified in the Methods section that this study should be regarded as an exploratory, pilot validation.
The study protocol was approved by the Riga Stradins University Research Ethics Committee on 21 March 2023 (Approval No. 2-PÄ’K-4/294/2023). All participants and their legal guardians (for minors) provided written informed consent prior to participation. This information was added to the Declaration section (lines 449 – 452).
Comment 8: Intervention: 1) Describe the sequence following the actual time sequence, starting with the warm-up, then the exercises. 2) Why did you not randomize the sequence of the two exercises as well as legs during the first exercise? If you can collect more data (for example from another 10 participants), change the sequence of tasks and legs. Then you solve two problems: Missing sequence randomization and too small sample size.
Reply 8: We thank the reviewer for these comments. The description of the sequence is corrected now (lines seque186 onward). We agree that randomising the sequence of exercises and leg conditions would be the optimal design in a full-scale validation study. Nevertheless, we deliberately chose a standardised sequence that closely mirrors how FIFA 11+–style injury-prevention tasks are implemented in practice. THe comment added in lines 201-203
Comment 9: Instrumentation: 1) Add the measurement/sensitivity range of the sensors used in the socks. 2) Clarify the sensor type. 3) Clarify how the socks are designed to make sure users are wearing the correctly (with the sensors at the right location). 4) Detailed information about the “entry-level force platform” is missing. Since it seems to be none of the commonly used systems that would be considered gold standard, you should provide technical details in addition to validation data of the platform itself (measurement accuracy).
Reply 9: We thank the reviewer for the constructive suggestions and have revised the manuscript accordingly.
The knitted conductive plantar-pressure sensors integrated into the socks exhibit a sensitivity in the range of approximately 0.5–3 μS/kPa. The variability within this range is primarily related to the pre-strain of the knitted structure, which can differ slightly depending on how the sock conforms to an individual foot. This information has been added to the revised manuscript.
We added a clearer description of the sensor type and their operating principle, together with the appropriate reference:
Oks, A., Katashev, A., Zadinans, M., Rancans, M., & Litvak, J. (2016). Development of smart sock system for gait analysis and foot pressure control. IFMBE Proceedings, 57, 466–469.
The corresponding details are now provided in lines 217–227.
We clarified how correct placement of the sensors on the plantar surface was ensured. The position of sensors was visually inspected and and manually adjusted to ensure that the forefoot sensors were placed beneath the metatarsal heads and the heel sensor along the midline of the calcaneus. This procedure was applied consistently for all participants and is now described in the manuscript, lines 237-239
Force platform details.
We expanded the description of the “entry-level force platform” to justify its role as the gold standard used in this study. The revised text now includes:– the manufacturer and model;
– the sampling frequency, resolution, and measurement range; – accuracy specifications; and – references to validation studies demonstrating its suitability for CoP assessment.
These details are now included in the Instrumentation section.
Comment 10: Analysis: 1) The procedure (from raw data to final data point for analysis) is unclear. After measurement you have a data time series throughout the exercise across multiple repetitions. Each frame has a different fractal CoP value. How many values did you use for the analysis (the entire time series, one CoPx per participant, or one CoPx per repetition), and – if not the entire time series – how did you derive those values (e.g., the average across frames) or – if the entire time series – how did you combine time series data from multiple participants? Based on the results, I can only assume that you tested the entire time series for each participant separately. It must be clearly described what you did. 2) Pearson correlation is not a suitable test for the current type of validation study. Pearson tests linear relationship. You need to test exact agreement. You need to conduct tests like ICC (be careful about the specification because some test consistency instead of exact agreement) or CCC (tests always exact agreement). In addition, you should calculate descriptive information such as RMSE and Bland-Altman’s limits of agreement.
Reply 10: The procedure of COP calculation was clarified – indeed the waveforms for X and Y coordinates of COP were calculated for each participant, each exercise, and each repetition (lines 276-277)
Following your recommendation, we replaced the Pearson correlation coefficient with the Concordance Correlation Coefficient (CCC). Alongside, we calculated RMSE and analysed Bland-Altman’s plots and estimated the percentahe of waveform point that fall outside the Bland- Altman limits.
The subsection 2.5 “Data analysis method” was thoroughly rewritten. The detailed description of used waveforms and metrics (Time-series plots, Correlation diagrams, Bland–Altman plots, The root mean square error (RMSE), Lin’s Concordance Correlation Coefficient) were described
Comment 11: Other aspects in this chapter are irrelevant as Pearson’s correlation needs to be replaced by proper statistical tests. However, generally there are the following problems:
Reply 11: The “results” section was nearly totally rewritten.
3) L197-198: The highest p value observed is a result and does not belong to the methods section. Just describe the significance level you used without results.
Agree. moved in the result section
4) L198-199: r = 0.05 is NOT the critical value for non-significance. The correlation coefficient r expresses the effect size of the correlation, not its significance, which is expressed by p.
The significance level alpha was meant. It is corrected now
5) L199-201: Unclear what that means, but it seems to be a result or even an interpretation (-> discussion section), so just remove.
Removed.
6) L201-205: Not necessary to explain the range of Pearson’s correlation coefficients (that is undergraduate-level knowledge, first statistics course). A reference for the interpretation benchmarks is missing.
The Table 1 with interpretation of the values of concordance correlation n coefficient was added, together with references, explaining selected limits .
Reviewer 2 Report
Comments and Suggestions for Authors
- Without seeing figure 1, I had no idea if "squat with heel raise" involves single or double legs. Suggest clarification in the abstract and in the introduction section.
- What is the reason that the participants were not tested wearing their football shoes? Football players don't play with their socks only, and wearing shoes can potentially alter the results. This decision should be further explained and considered a limitation of the study.
- Introduction: Chronic ankle instability is also common in football players. It should be included and discussed.
- Introduction: Please avoid paragraphs with very few sentences: line 101-110.
- Method: The reliability of DAid Smart Socks?
- Discussion: The section should be broken down into several paragraphs.
- Limitation: The 2 activities chosen were not as dynamic as real-life football activities in the field that involve speed and agility. More challenging activities, such as the Side Hop Test or Figure of Eight Hop Test, should be considered in future studies.
Author Response
Thank you so much for your comments!
Comment 1: Without seeing figure 1, I had no idea if "squat with heel raise" involves single or double legs. Suggest clarification in the abstract and in the introduction section.
Reply 1: Thank you for this helpful suggestion. We agree that the distinction between unilateral and bilateral execution was not sufficiently clear. To improve clarity, we revised the Abstract and Introduction to explicitly state that the single-leg squat is a unilateral task and that the squat with heel raise is performed bilaterally. These changes ensure that readers understand the movement characteristics even before viewing Figure 1.
Comment 2: What is the reason that the participants were not tested wearing their football shoes? Football players don't play with their socks only, and wearing shoes can potentially alter the results. This decision should be further explained and considered a limitation of the study.
Reply 2: Thank you for this important observation. We agree that footwear influences plantar pressure and CoP characteristics. We have now added a clear explanation to the Methods section describing why the tests were performed without shoes—primarily to ensure consistent sensor-surface contact and to avoid footwear-related variability during validation against the force platform. Additionally, we expanded the Limitations section to acknowledge that testing without football footwear restricts the generalisability of the results to real match conditions. These revisions were added in the Methods section 170 - 177 and in the Limitations section 379-385.
Comment 3: Introduction: Chronic ankle instability is also common in football players. It should be included and discussed.
Reply 3: Thank you for this valuable point. We have now expanded the Introduction to include chronic ankle instability (CAI) as an important condition among football players. The revised text highlights its relevance to altered plantar loading, neuromuscular deficits, and increased injury risk. This addition strengthens the rationale for assessing CoP behaviour in youth players. The new sentences were added 70 - 73 in the Introduction.
Comment 4: Introduction: Please avoid paragraphs with very few sentences: line 101-110.
Reply 4: We appreciate the reviewer’s suggestion to improve the readability of the Introduction. The short paragraph in lines 101–110 has now been merged with and expanded within the preceding content to provide a more coherent and continuous discussion of existing evidence on smart garments and DAid® Smart Socks. The paragraph has been rewritten to present a fuller, better-structured rationale for the study.
Comment 5: Method: The reliability of DAid Smart Socks?
Reply 5: Thank you for this suggestion. We have now added a dedicated explanation of the existing reliability evidence for DAid® Smart Socks within the Instrumentation section. Specifically, we reference findings by Eizentals et al. (2021) [15], who reported moderate-to-excellent ICC values when comparing repeated plantar pressure measurements against the Pedar® system, and we also cite related reliability work on DAid® textile systems (Semjonova et al., 2019) [17]. These additions clarify that the system demonstrates stable measurement behaviour and acceptable reliability.
Comment 6: Discussion: The section should be broken down into several paragraphs.
Reply 6: Thank you for this helpful suggestion. We have revised the Discussion section by dividing the previously long, continuous text into multiple thematically structured paragraphs. These breaks improve clarity and readability by separating:
(1) Study aim + summary of main findings;
(2) Literature on prior DAid® validation studies;
(3) Implications for sports medicine and neuromuscular control: relationship between CoP behaviour, injury risk, and movement quality;
(4) Technical limitations of textile sensors;
(5) Future research direction.
Comment 7: Limitation: The 2 activities chosen were not as dynamic as real-life football activities in the field that involve speed and agility. More challenging activities, such as the Side Hop Test or Figure of Eight Hop Test, should be considered in future studies.
Reply 7: We agree with the reviewer that the selected exercises reflect controlled neuromuscular tasks rather than high-intensity football-specific movements. To address this, we have expanded the Limitations section to acknowledge that the tasks do not represent the speed, agility, and multidirectional loading encountered in actual football settings. We now explicitly state that future studies should incorporate more dynamic tests, such as the Side Hop Test or Figure-of-Eight Hop Test, to evaluate system performance under more demanding and realistic conditions. This addition was inserted in the Limitations section.
Reviewer 3 Report
Comments and Suggestions for Authors
This study evaluates the effectiveness of DAid® Smart Socks in the context of football performance monitoring and concludes that they represent a highly cost-effective solution. The structure and organization are clear. Nonetheless, several concerns and areas requiring further refinement have been identified:
1. Given that DAid® Smart Socks incorporate only six sensors, the sensor configuration is relatively sparse. This raises questions regarding how inter-individual variations in foot morphology may influence the accuracy and reliability of the measurement outcomes.
2. The integration of data from Figures 5 and 6 into a single visualization obscures the temporal evolution of measurement errors. It is therefore recommended to present these results using a three-dimensional spatiotemporal graph to enhance clarity and facilitate interpretation.
3. In certain instances, the observed test errors are notably large. Clarification is needed on the underlying factors contributing to such discrepancies, including potential sources related to sensor placement, movement artifacts, or signal processing limitations.
4. It would be valuable to investigate whether motion pattern recognition can be leveraged to dynamically correct sensor errors or to estimate system confidence levels under varying activity conditions. Additionally, the implementation of calibration strategies or adaptive filtering techniques could serve as potential approaches to mitigate measurement inaccuracies.
5. A comparative analysis with state-of-the-art smart sock technologies is currently lacking. To better position the proposed system within the existing landscape, a discussion on the measurement precision achieved in recent literature and benchmarking against commercially available solutions is warranted.
6. Information regarding the commercial pricing and manufacturing cost of DAid® Smart Socks remains unspecified. Providing such details would strengthen the assessment of its economic feasibility and scalability for broader adoption.
Author Response
Thank you so much for your comments and suggestions.
Comments 1: Given that DAid® Smart Socks incorporate only six sensors, the sensor configuration is relatively sparse. This raises questions regarding how inter-individual variations in foot morphology may influence the accuracy and reliability of the measurement outcomes.
Response 1: Thank you for this important observation. We fully agree that the six-sensor configuration represents a relatively sparse sensing layout and may therefore be influenced by inter-individual differences in foot morphology. This consideration is explicitly incorporated into the study aim, as we set out not only to evaluate the preliminary validity of the DAid® Smart Socks but also to examine how the magnitude and consistency of agreement with the force platform vary across individual participants. The Results and Discussion sections address this variability and its implications for system performance.
Comments 2: The integration of data from Figures 5 and 6 into a single visualization obscures the temporal evolution of measurement errors. It is therefore recommended to present these results using a three-dimensional spatiotemporal graph to enhance clarity and facilitate interpretation.
Response 2: We appreciate the reviewer’s suggestion to use a three-dimensional spatiotemporal graph. We explored this option during manuscript preparation; however, the 3D representation resulted in a visually cluttered figure that was more difficult to interpret than the 2D plots. Our primary objective with Figure 6 was to clearly demonstrate the absence of outliers and the consistency of CoP coordinate values across recordings. For this purpose, the chosen 2D visualization provided better readability and more effectively communicated the intended message.
Comments 3. In certain instances, the observed test errors are notably large. Clarification is needed on the underlying factors contributing to such discrepancies, including potential sources related to sensor placement, movement artifacts, or signal processing limitations.
Response 3: Thank you for highlighting this point. Larger test errors are indeed observed in some recordings, and these discrepancies are primarily attributable to factors inherent to textile-based sensing. Specifically, the sensitivity of the knitted sensors can vary considerably depending on the degree of stretch applied to the sock during put-on, as well as inter-individual differences in foot size and anatomy that influence sensor-to-skin contact and local loading conditions. One of the aims of our study was precisely to evaluate how often such suboptimal performance occurs and to identify the factors contributing to it. The manuscript discusses these sources of variability and their impact on measurement accuracy.
Comments 4. It would be valuable to investigate whether motion pattern recognition can be leveraged to dynamically correct sensor errors or to estimate system confidence levels under varying activity conditions. Additionally, the implementation of calibration strategies or adaptive filtering techniques could serve as potential approaches to mitigate measurement inaccuracies.
Response 4: We thank the reviewer for this insightful and forward-looking suggestion. The potential use of motion pattern recognition, dynamic confidence estimation, and adaptive calibration or filtering strategies indeed represents a promising direction for enhancing measurement robustness in textile-based sensing systems. We have now acknowledged this point in the Discussion as a potential avenue for future development and optimization of the DAid® Smart Socks.
Comments 5. A comparative analysis with state-of-the-art smart sock technologies is currently lacking. To better position the proposed system within the existing landscape, a discussion on the measurement precision achieved in recent literature and benchmarking against commercially available solutions is warranted.
Response 5: We appreciate the reviewer’s suggestion to include a comparative perspective with other smart sock technologies. Many commercially available systems rely on rigid or semi-rigid pressure sensors, advanced multi-sensor arrays, or proprietary manufacturing methods, which contribute to higher production costs and limit accessibility. In contrast, the DAid® Smart Socks use a textile-integrated sensor design that can be produced on standard commercial knitting machines, making the system substantially more cost-efficient and comparable in price to ordinary sports socks. This technological peculiarity positions the DAid® system differently within the current landscape—prioritizing affordability, scalability, and wearability rather than high-density sensor configurations. We have now added a brief discussion of this positioning in the manuscript to clarify how the proposed system compares with existing solutions.
Comments 6. Information regarding the commercial pricing and manufacturing cost of DAid® Smart Socks remains unspecified. Providing such details would strengthen the assessment of its economic feasibility and scalability for broader adoption.
Response 6: We thank the reviewer for raising the question of commercial pricing and manufacturing cost. At this stage, detailed cost information is considered commercially confidential and cannot be disclosed within the manuscript. However, we would like to note that the DAid® Smart Socks are designed to be produced using standard knitting machines, which supports cost-efficient manufacturing and potential scalability. A full economic analysis, including specific pricing, falls outside the scope of the present validation study.
Round 2
Reviewer 3 Report
Comments and Suggestions for Authors
The overall quality of the manuscript has been substantially enhanced, and it is now considered suitable for publication.
Author Response
We sincerely thank the reviewers for their positive evaluation of the revised manuscript and for their constructive comments, which have helped to substantially improve the quality of the paper.